# The Activation of Cytochrome P450 2C9 Is Facilitated by the Coenzyme Forms of Vitamin B2

**DOI:** 10.3390/molecules30183673

**Published:** 2025-09-10

**Authors:** Polina I. Koroleva, Alexey V. Kuzikov, Andrei A. Gilep, Sergey V. Ivanov, Alexander I. Archakov, Victoria V. Shumyantseva

**Affiliations:** 1Institute of Biomedical Chemistry, Pogodinskaya Street, 10, Build 8, 119121 Moscow, Russia; 2Faculty of Biomedicine, Pirogov Russian National Research Medical University, Ostrovitianov Street, 1, 117997 Moscow, Russia; 3Institute of Bioorganic Chemistry, 220084 Minsk, Belarus

**Keywords:** CYP2C9, electrochemical analysis, diclofenac, naproxen, riboflavin, electron transfer, enzymatic assay, protective properties

## Abstract

The activity of cytochrome P450 enzymes decreases in older adults, which can lead to toxic effects from polypharmacy. Cytochromes P450 are the most significant enzymes involved in the metabolism of foreign compounds, including pharmaceutical substances. Vitamin B2, or riboflavin (RF), is a potent antioxidant that is vital for the body and participates in numerous enzyme-catalyzed redox reactions. RF is phosphorylated intracellularly to form flavin mononucleotide (FMN), which is further metabolized into flavin adenine dinucleotide (FAD). The active site of the NADPH-dependent cytochrome P450 reductase (CPR), a redox partner of CYP enzymes, is necessary for the catalytic functions of cytochromes P450. The active site of reductase is a complex formed by two types of vitamin B2, such as flavin adenine dinucleotide (FAD) and flavin mononucleotide (FMN). In our study, we investigated the impact of the phosphorylated form of vitamin B2, FAD, and FMN on the catalytic activity of cytochrome P450 2C9 (CYP2C9) towards non-steroidal anti-inflammatory medications diclofenac and naproxen. It was shown that FAD significantly enhanced the catalytic efficiency of CYP2C9. The 4-hydroxylation of diclofenac was enhanced by 148 ± 10%. The O-demethylation of naproxen showed an increase of 120 ± 14%. Based on these data, we can assume that intake of vitamin B2 (riboflavin) improves catalytic efficiency of CYP2C9. This finding is essential for the modulation of catalytic activity of CYP2C9. The proposed electroanalytic approach is a sensitive and robust method for drug metabolism assay.

## 1. Introduction

At present, 8% of the global population is aged 65 and older, a proportion projected to rise to 16% by 2050 and 29.5% in 2060 [1,2]. The main consumers of medications are elderly people, due to their chronic diseases and general ailments. Older people are the main users of many drugs (i.e., polypharmacy). It is well known that elderly people used vitamins for improving the quality of life and for the prevention or delay of the onset of age-related health deterioration [3]. B2 deficiency can lead to anemia, inflammation, cognitive impairment, depression, and, during embryonic development, the risk of congenital malformations [4,5].

Unregulated medication intake, especially among older people, can cause undesirable side effects. First-pass metabolism mainly occurs in the liver by cytochromes P450 (CYPs), the most important enzymes of drug conversion. CYPs are the primary enzymes responsible for metabolizing exogenous compounds such as xenobiotics, drugs, and toxins, as well as synthesizing endogenous compounds like steroids, fatty acids, and vitamins. The activity of CYP enzymes decreases in older adults, which can lead to toxic effects of polypharmacy [1].

Human CYP enzymes account for 75–80% of all phase I drug metabolism [2,3]. The unique properties of cytochrome P450s as biocatalysts include their high diversity in the metabolism of both endobiotics and xenobiotics. Activation or inhibition of catalytic activity of CYP enzymes is a very important medicinal aspect of pharmacology and, especially, human health and anti-aging [6]. Another side of this problem is biotechnology CYP applications, based on the unique properties of CYP enzymes such as regio- and stereo-selectivity, as well as their broad substrate spectrum. These peculiarities have led to extensive efforts to influence P450 systems in order to overcome the inherent limitations of native enzymes [2,3]. Earlier we showed that the catalytic activity of CYP3A4 and CYP2C9 isoforms may be regulated by metabolic antioxidant medication, such as taurine, mexidol, glutathione, and vitamins with antioxidant properties [7,8]. Antioxidants can diminish the level of reactive oxygen species (ROS) by radical-scavenging effects or modulating the activities of CYPs, which are known to generate reactive intermediates during catalysis [3,9,10].

The main feature of CYP enzymes is their complex and multistage catalytic cycle, which involves the participation of electrons from NADPH, an electron donor, through a protein redox chain [9,10]. The catalytic activity of cytochromes P450 directly depends on the presence of flavin nucleotides in the body, since their work requires CPR containing flavin cofactors in the active center. The relationship between vitamin deficiency and the activity of enzymes involved in the biotransformation of drugs has been studied [11]. Riboflavin (RF, vitamin B2) is a water-soluble member of the B-vitamin family with antioxidant, anti-aging, anti-inflammatory, anti-nociceptive, and anti-cancer properties [12,13,14,15]. RF, chemically, is 7,8-dimethyl-10-ribityl-isoalloxazine, which consists of a flavin isoalloxazine ring bound to a sugar side chain, ribitol [14]. Riboflavin is converted in the body into its coenzyme form by the action of riboflavin kinase. This enzyme converts riboflavin into FMN, which is then converted to FAD by FMN adenylyltransferase (Figure 1) [15].

Riboflavin (vitamin B2) is one of the most important compounds obtained by humans from food [15]. The role of flavins in the body is diverse; for example, as a cofactor of important drug-metabolizing enzymes, flavin monooxygenases, and redox-partner protein of cytochrome P450 NADPH-dependent cytochrome P450 reductase (CPR). Flavins participate in the electron transfer chain and as cofactors of dehydrogenases—an important part of catabolic processes. Riboflavin plays a critical role in the body’s antioxidant defense mechanism as a constituent of glutathione reductase [12]. Furthermore, the impact of riboflavin on superoxide dismutase and catalase activity has been demonstrated [13]. There is a decrease in the level of vitamin B2 in the elderly due to an insufficient amount of riboflavin in the diet [14]. In addition, as it was shown in [16], there is a lack between consumption of riboflavin and its biochemical deficiency, which could be explained by the decrease in the absorption of B vitamins in the gastrointestinal tract in the elderly. Insufficient intake of riboflavin encourages the next disorder, as chapped lips, sore throat, and skin diseases, as well as depressive states, are associated with a deficiency of other vitamin B groups [14,16]. The intake of riboflavin helps to mitigate the risk of developing osteoporosis in females [17]. The study conducted by Wang et al. [18] found a positive correlation between dry mouth and the risk of overall mortality in elderly individuals. The intake of vitamin B2 at sufficient levels was associated with a decreased risk of mortality among participants experiencing dry mouth. RF intake reduces the risk of cataract and osteoporosis, reduces migraine, boosts the immune system, and demonstrates a protective effect on medical conditions such as sepsis and ischemia [14,15,16,17,18].

Cytochrome P450 2C9 (CYP2C9) is one of the most significant human cytochrome P450 enzymes and the second most abundant cytochrome expressed in the liver. It comprises approximately 20% of all P450 isoforms in the organ, catalyzing approximately 15–20% of drug metabolism. This enzyme is particularly important for drugs with a low therapeutic index, as it plays a significant role in their metabolism. CYP2C9 plays a role in the metabolism of various drugs, including acenocoumarol, candesartan, celecoxib, fluvastatin, glyburide, ketamine, methadone, phenytoin, tolbutamide, phenobarbital, S-warfarin, piroxicam, losartan, tamoxifen, and non-steroidal anti-inflammatory medications, such as diclofenac, ibuprofen, and naproxen [3,4,19]. Furthermore, CYP2C9 is responsible for the metabolism of endogenous compounds, including steroid hormones, vitamin A, and fatty acids. Additionally, the CYP2C9 enzyme exhibits polymorphism, which can introduce further challenges in prescribing medications that are substrates or activators of this enzyme [19,20].

Electrochemical systems that involve direct electron transfer between enzymes and electrodes are useful not only in gaining insight into electrocatalytic mechanisms but also in developing efficient assay systems for investigating drug–drug interactions, biocatalysis, screening for potential substrates or inhibitors, and designing bioreactors [21]. The use of electrochemical catalysis, where the electrode acts as the electron donor, enables the creation of efficient cytochrome P450 catalytic systems [22]. Improving the efficiency of these systems is crucial for screening the potential of cytochrome P450 substrates and inhibitors and investigating drug interactions, as this allows for more sensitive registration and investigation of catalytic processes. Furthermore, these productive electrocatalytic systems have applications in the development of bioreactors and the detoxification of toxic organic compounds that are substrates for hemoproteins involved in remediation processes [23].

Previously, it has been shown that flavins can serve as effective models of flavoprotein reductases CPR [24,25]. The interactions between cytochrome P450 2B4 and riboflavin, which acts as a surrogate for flavoproteins, have been studied both covalently and non-covalently [26,27,28]. The heme region of self-sustained flavohemoproteins such as BM3 from Bacillus megaterium exhibits catalytic activity in the presence of NADPH and FMN [29]. In the presence of NADPH, bacterial cytochromes like CYP106A2, CYP107DY1, and HmtS catalyze the reaction of N-dealkylation of diphenhydramine, with riboflavin serving as a mediator [30]. Cytochromes P450 3A4 and the reductase domain of self-sufficient Bacillus megaterium cytochrome BM3 have also been studied as part of a catalytic system [29].

Riboflavin, FMN, and FAD, which are low molecular weight substituents of NADPH-dependent cytochrome P450 reductase, have been shown to be effective additives in enhancing the efficiency of cytochrome P450-mediated electrocatalysis using CYP3A4 as the main enzyme for drug/xenobiotic metabolism [31]. Flavins play a dual role in the catalysis of CYP enzymes. They not only served as a reductase model but also played the role of antioxidant molecule [14].

FAD has been used in this study to optimize the electron transfer pathway in electrochemically driven CYP2C9 catalysis, with the goal of increasing the electrocatalytic activity of CYP2C9. A previously established method based on differences in oxidation potentials between the substrate (diclofenac) and the product (4′-hydroxylated diclofenac), a metabolite of CYP2C9, was employed to analyze the produced 4′-hydroxy-diclofenac as a product of enzyme reaction [32]. Thus, the study aimed to assess the effect of vitamin B2 on the metabolic conversion of two non-steroidal anti-inflammatory drugs (NSAIDs), diclofenac and naproxen, by the CYP2C9 enzyme.

## 2. Results and Discussion

### 2.1. Spectrophotometric Analysis of CYP2C9 and FAD Complexation

Absorption spectra of the complex of CYP2C9 + FAD revealed a Soret peak characteristic of heme proteins (400–416 nm) and a typical shoulder at 450–475 nm, indicating the presence of flavins, as depicted in Figure 1A, with a concentration dependence on the flavin content (Figure 1B). The complex of CYP2C9 + FAD demonstrates a hyperchromic effect with a bathochromic shift (from 416 nm to 419 nm), confirming the interaction of these components. These results are comparable to those obtained in [28].

The value of the binding constant for CYP2C9 and FAD is Kd = 174 ± 52 µM, which is consistent with the data presented in the work, where the association constant between CYP2B4 and riboflavin was 18,000 M^−1^ (Kd = 56 µM) [28]. We suggest that the adenine moiety enhances the binding constant of FAD in comparison with RF. Probably the affinity properties of CYP2C9 may differ from the CYP2B4 enzyme.

### 2.2. Characterization of the CYP2C9 Electrochemical System

Cytochrome P450 as an enzymatic biocatalyst requires redox-partner proteins (NADPH-dependent cytochrome P450 reductase (CPR) and cytochrome b5). Reductase is a flavoprotein containing both FAD and FMN as prosthetic groups. NADPH supplies electrons for FAD and FMN with subsequent transfer of electrons to heme iron of cytochrome P450 [23,33,34]. Flavin nucleotides play many roles, such as the regulation of the flow of electrons, increasing the coupling of hydroxylation of substrates, and influencing the conformational changes in the structure of the protein [7]. For the optimization of the electron transfer chain in cytochrome P450 3A4, we used riboflavin, FMN, and FAD as substitutes of reductase flavoprotein for the enhancement/improvement of electron transfer in electrochemically driven cytochrome P450 catalysis [25]. The electron transfer pathway in the CYP+flavin complex occurs in accordance with the scheme: NADPH or electrons from electrode → flavin(s) → heme (Fe^+3^) [25,26,27,28,29,34,35]. Earlier we showed that the most electroactive complex was CYP3A4 + FAD [31], demonstrating the highest electroactive surface coverage.

Investigation of electrochemical properties of CYP2C9 immobilized on a screen-printed electrode modified by DDAB was described in detail for aerobic and anaerobic conditions earlier [20]. In an argon-saturated buffer, a screen-printed electrode modified with DDAB and immobilized cytochrome P450 2C9 (SPE/DDAB/CYP2C9) demonstrated two peaks at Eox = −0.252 ± 0.006 V as an oxidation anodic peak and E_red_ = −0.383 ± 0.01 V as a reduction cathodic peak using cyclic voltammetry (CV) (all potentials were referred to a Ag/AgCl reference electrode). For the immobilized CYP2C9 + FAD complex (with a 1:1 ratio), Eox = −0.235 ± 0.012 V and E_red_ = −0.401 ± 0.001 V were registered, attributed to the reversible redox reaction of heme iron in accordance with a scheme Fe (III) + e ↔ Fe(II) (Figure 2A). The midpoint potentials E^0^′ were calculated from the values of anodic and cathodic peak potentials by the equation E^0^′ = (E_red_ + E_ox_)/2. Cytochrome CYP2C9 and CYP2C9 + FAD demonstrated close midpoint potential values as E^0^′ = −0.318 ± 0.03 V and −0.319 ± 0.013, respectively (Table 1).

The linear relationship between the cathodic and anodic current peaks and the scan rate supports the mechanism of a surface-controlled electron transfer process between CYP2C9 and the SPE/DDAB electrode, which is known as protein film voltammetry (Figure 2B).

We did not observe the electroactivity of the FAD molecule on the SPE/DDAB (Figure 2C, black line). However, the CV of SPE/DDAB/CYP2C9 + FAD in comparison with the SPE/DDAB/CYP2C9 demonstrates more pronounced peaks in the CV curves, confirming the role of FAD as an effective electron transfer mediator [31]. Figure 2D shows a trumpet plot for the response of SPE/DDAB/CYP2C9 + FAD. The trumpet plot allows us to verify the range of scan rates for the surface-controlled electron transport process between the CYP2C9 and the SPE/DDAB [36]. Reductive (red circle) and oxidative (black circle) peak potentials were plotted against the logarithmic scale of the scan rate. A linear regression was plotted with a coefficient of determination R2 = 0.96. At scan rates higher than 100 mV/s, a diffusion-controlled mechanism for electron transfer can be observed. The ratio of cathodic to anodic peak currents, Ic/Ia, corresponds to a quasi-reversible electrochemical response, with a value of 0.96 [37].

The difference (ΔEp) between the anodic peak potential (Eox) and the cathodic peak potential (E_red_) calculated for the CYP2C9 + FAD complex embedded in DDAB at the scan rate of 0.10 V/s was found to be 0.168 ± 0.006 V. This value exceeds the expected theoretical value of 0.059 mV [37,38], indicating a deviation from the expected behavior [38]. The deviation of the theoretical value may indicate the influence of FAD on the electron transfer characteristics of heme.

The kinetics of heterogeneous electron transfer was analyzed using the Laviron model [39]. The heterogeneous electron transfer constant (*k_s_*) values for the electron transfer between the CYP enzyme and SPE/DDAB at the scan rate of 0.1 V s^−1^ were determined using Equation (1):(1)$$k_{s}=\frac{mFnv}{RT}\;$$
where *k_s_* is the heterogeneous electron transfer constant s^−1^; *m* is not a value that can be determined experimentally, and it depends on E^0^′; in this particular case, the value of 1/m is equal to 9, because E^0^′ = 0.168 ± 0.006 (this value was obtained from reference [35]); *F* is the Faraday constant, 96485 C mol^−1^; *ν* is the scan rate, V·s^−1^; *n* is the number of electrons transferred (*n* = 1 for cytochrome P450s); *R* is the gas constant, 8.314 J mol^−1^ K^−1^; and *T* is the temperature, K.

The electrochemical parameters derived from the experimental data for the SPE/DDAB/CYP2C9 system compared to the SPE/DDAB/CYP2C9 + FAD system in an argon-saturated electrolyte buffer are presented in Table 1.

The electroactive surface concentration Γ_0_ of CYP2C9 on the electrode was determined from the cyclic voltammetry in an argon-saturated, anaerobic buffer solution. The amount of electroactive protein was calculated according to Faraday’s law by Equation (2):(2)$$\Gamma_{0}=\frac{Q}{nFA}\;$$
where Γ_0_ is the surface coverage or surface concentration of electroactive protein, mol cm^−2^; *Q* is the electric charge calculated from integration of voltammetry peaks, C; n and *F* values are provided above; and *A* is the surface area of the working electrode, cm^2^ [40]. The integration of reduction peak area in the anaerobic cyclic voltammograms yielded the charge in coulombs.

Based on the analysis of the anodic and cathodic peak surface areas, the electroactive protein coverage (Γ_0_) was calculated in mol/cm^2^. The Γ_0_ value for CYP2C9 + FAD was found to be greater than 40%, as compared to that of CYP2C9 (see Table 1). This difference in the electroactive area confirms the assumption that there is an increased efficiency of electron transfer between the enzyme’s active site and the electrode surface.

### 2.3. The Influence of Coenzyme Forms of Vitamin B2 on the O-Demethylase and Hydroxylase Activities of the CYP2C9 Enzyme on Non-Steroidal Anti-Inflammatory Drugs, Naproxen and Diclofenac

Nonsteroidal anti-inflammatory drugs are used in the treatment of pain, inflammation, and fever and can be used to relieve chronic symptoms and to relieve acute conditions. Prolonged use of nonsteroidal anti-inflammatory drugs can cause a number of side effects, such as blood clotting disorders and gastric and duodenal ulcers, especially when their metabolism is disrupted by insufficient cytochrome P450s activity.

Diclofenac is the sodium salt of 2-[2-(2,6-dichloroanilino)-phenyl]acetic acid, which is a derivative of phenylacetic acid. Diclofenac is a non-steroidal anti-inflammatory, antipyretic, and anti-cancer drug [41,42,43,44,45]. The metabolism transformations of diclofenac by CYP enzymes are stereoselective. CYP3A4 catalyzes hydroxylation of diclofenac to form 5′-hydroxydiclofenac, while CYP2C9 catalyzes formation of the 4′-hydroxydiclofenac (Figure 2) [46].

Our study aimed to investigate the effect of coenzyme forms of vitamin B2 in complexes with CYP2C9 on the efficiency of drug metabolism. Cyclic voltammetry was used to evaluate the electrochemical parameters of comparative catalytic processes of CYP2C9 or CYP2C9 + FAD immobilized on SPE/DDAB when interacting with diclofenac or naproxen.

The background-subtracted cyclic voltammograms showed the broad reduction non-catalytic peak centered at E_red_ = −0.425 V for CYP2C9 and at E_red_= −0.420 V for the CYP2C9 + FAD complex (Figure 3). Reduction catalytic peak Ecat with potential shifts of 0.02 V was observed in the presence of diclofenac (Table 2). However, the CYP2C9 + FMN complex revealed E_red_ = −0.418 ± 0.005 V, which was more negative in comparison with CYP2C9 and CYP2C9 + FAD, reflecting a thermodynamically less effective process. A higher ratio (I_cat_/I_red_) between the catalytic current in the presence of diclofenac I_cat_ and the reduction current I_red_ in the presence of oxygen for the complex of CYP2C9 + FAD and CYP2C9 + FMN confirms the effectiveness of the electrocatalytic process of CYP2C9 + FAD-mediated diclofenac hydroxylation (Table 2).

The ratio I_cat_/I_red_, named as the coupling index, reflects the repartition of the electron flow between substrate and oxygen and can serve as an indicator of coupling efficiency in the catalysis of cytochrome P450s [20]. The analytical sensitivity of the bio-electrode SPE/DDAB/CYP2C9 to diclofenac as substrate is 0.0045 A/M (S/N = 3), and SPE/DDAB/CYP2C9 + FAD is 0.0065 A/M. The potentials for the start of catalysis, E_onset_ of diclofenac for the CYP2C9 and for complex with vitamin B2 derivatives, are shifted to the anodic positive region by approximately 0.2 V compared to the potential of catalysis, E_cat_ (Table 2 and Figure 3).

Previously, we had developed a two-electrode system for the estimation of metabolites of CYP-dependent catalysis. Our all-electrochemical approach is based on the estimation of the electrochemical oxidation of a metabolic product using an SPE modified with single-walled carbon nanotubes (SPE/SWCNT) [41]. Electroenzymatic efficiencies of CYP2C9, CYP2C9 + FMN, and CYP2C9 + FAD were determined electrochemically by registering 4′-hydroxydiclofenac (Figure 4). Differences in electrochemical oxidation potentials of 4′-hydroxydiclofenac (E = +0.12 V) and diclofenac (E = +0.25, +0.5, and +0.85 V) on the square-wave voltammograms (SWVs) allows to detect and calculate the quantity of product after enzymatic electrolysis [32].

Based on the results obtained, we determined that the V_max_ value for CYP2C9-dependent diclofenac 4′-hydroxylation in the electrochemical system is equal to 1.89 ± 0.21 × 10^−9^ M/min, 2.14 ± 0.16 × 10^−9^ M/min, and 2.80 ± 0.29 × 10^−9^ M/min for CYP2C9, CYP2C9 + FMN, and CYP2C9 + FAD, respectively (Table 3).

To assess the comparable kinetic parameters of the electrochemical systems, we performed amperometric titration of CYP2C9 + FMN and CYP2C9 + FAD immobilized on SPE/DDAB with diclofenac. The apparent Michaelis constants, K_M_, were calculated from electrochemical data using the electrochemical form of the Michaelis–Menten equation with Lineweaver–Burk linearization (Equation (3)). The amperometric and the Lineweaver–Burk diagram in double reciprocal coordinates of the dependence of the catalytic current of SPE/DDAB/CYP2C9 + FMN and SPE/DDAB/CYP2C9 + FAD electrodes for experimental amperometric titration of diclofenac are shown in Figure 5A–D.(3)$$\frac{1}{I_{cat}}=\frac{K_{M}^{app}}{I_{cat\;max}\left[S\right]}+\frac{1}{I_{cat\;max}}$$

Based on the results of titration in double reciprocal coordinates (Lineweaver–Burk diagram in 1/V, 1/[S]), the values of the Michaelis constants, K_M_, were calculated and summarized in Table 3.

As can be seen from Table 3, the complex formation of CYP2C9 with FAD significantly improved the catalytic properties of the system. Despite the decrease in the Michaelis constant, K_M_, relative activity, measured as 4′-hydroxydiclofenac formed, increased and corresponded to 148 ± 10% in comparison with CYP2C9.

Naproxen is a non-steroidal anti-inflammatory medication metabolized by CYP2C9 (Figure 3) [42].

Naproxen is an inhibitor of cyclooxygenase 1 and 2 and is used to relieve pain in various diseases. The comparative catalytic activity of CYP2C9 and CYP2C9 + FAD with naproxen as substrate was studied. The positive role of FAD in electroenzymatic O-demethylation of naproxen catalyzed by CYP2C9 was evidenced. There is the anodic shift in the potentials of catalysis Ecat from −0.432 V for the CYP2C9 to −0.322 V for CYP2C9 + FAD complex in the presence of 100 µM naproxen, reflecting a more thermodynamically favorable process. A significant increment in catalytic current Icat was also registered in comparison with CYP2C9 (Figure 6, Table 4).

The measurement of the catalytic performance of CYP2C9 in the electrochemical reaction of naproxen at a potential of E = −0.6 V for a duration of 60 min was conducted by quantifying the concentration of the reaction by-product, formaldehyde. The determination of formaldehyde was quantified spectrophotometrically by measuring the absorbance at 412 nm through the formation of a colored product during the Hantzsch reaction [44].

We observed an increase in the catalytic activity of the CYP2C9 + FAD complex when naproxen was used as a substrate. The catalytic current of naproxen (I_cat max_) for the CYP2C9 + FAD complex increased from −0.34 ± 0.07 µA to −4.220 ± 0.08 µA in comparison with CYP2C9 (Table 4).

FAD also enhanced the electroenzymatic efficiency up to 120%, with increasing V_max_ from (6.44 ± 1.17) × 10^−11^ M/min to (7.77 ± 1.15) × 10^−11^ M/min for the CYP2C9 and CYP2C9 + FAD complexes, respectively.

Earlier we investigated the effect of phospholipid nanoparticles on CYP2C9-mediated metabolism of naproxen [41]. It was shown that phospholipid nanoparticles enhance the catalytic efficiency of CYP2C9 to naproxen but decrease the electrochemical parameters of non-catalytic systems. FAD as a cofactor of reductase not only improved the catalytic efficiency of CYP2C9-mediated O-demethylation of naproxen but also elevated the electroanalytical parameters of CYP2C9, such as the surface coverage or surface concentration of electroactive protein.

Based on these results, we can assume that the intake of vitamin B2 as a precursor of NADPH-dependent cytochrome P450 reductase cofactors promotes the performance of CYP2C9 as a drug-metabolizing enzyme.

## 3. Materials and Methods

Human recombinant CYP (210 µM CYP2C9 stock solution in 550 mM potassium phosphate buffer (pH 7.2), containing 0.2% CHAPS, 1 mM dithiothreitol, and 20% glycerol (*v*/*v*)) was purified according to the protocol [45] in the Institute of Bioorganic Chemistry (Minsk, Belarus). Concentration of CYP2C9 was determined by a complex formation of the reduced form with carbon monoxide using the absorption coefficient ɛ_450_ = 91 mM^−1^cm^−1^ [46]. Chloroform (99.8%), didodecyldimethylammonium bromide (DDAB, 98%), disodium phosphate (≥99%), diclofenac sodium salt (≥98%), and 4-hydroxydiclofenac (≥98.5%) were obtained from Sigma-Aldrich (St. Louis, MO, USA). Flavin adenine dinucleotide (FAD, 99%) was obtained from Fluka (Buchs, Switzerland). Potassium hydroxide (98%), potassium dihydrogen phosphate (98%), and sodium chloride (98%) were purchased from Spectrchem (Moscow, Russia). Flavin mononucleotide (10 mg/mL) was obtained from PharmStandart (Moscow, Russia). Water dispersion of 0.2% single-wall carbon nanotubes (SWCNTs, diameter 1.6 ± 0.4 nm, length ≥ 5 μm, surface area 1000 m^2^/g) TUBALL™ BATT H_2_O stabilized by carboxymethylcellulose obtained from OCSIAL Ltd. (Novosibirsk, Russia).

All electrochemical experiments were carried out at room temperature (25 °C) and in 0.1 M potassium phosphate pH 7.4 containing 0.05 M NaCl using a potentiostat/galvanostat PGSTAT 302N Autolab (Metrohm Autolab, the Netherlands), controlled by NOVA software (version 2.0).

Screen-printed electrodes (SPEs), which consist of the graphite working electrode (surface area 0.0314 cm^2^), an auxiliary electrode, and a silver/silver chloride pseudo reference electrode, were used and obtained from ColorElectronics, Russia (Moscow, Russia).

To carry out electrode modification, 1 μL of 0.1 M DDAB in chloroform at the surface of the working electrode was applied, then electrodes were left for 10 min until completely dried. Immobilization of the enzyme was carried out as described earlier in [16], but in this case, we used 0.5 µL of 210 µM CYP2C9 solution. A molar ratio of 1:1 of CYP2C9 to FAD or FMN was used to form a non-covalent complex on the electrode surface. All potentials were referred to a silver/silver chloride reference electrode (Ag/AgCl). Enzymatic electrocatalytic reactions in the presence of substrate diclofenac under aerobic conditions were performed at room temperature in an air-saturated buffer; a potential of E = −0.6 V was applied for 90 min. Cyclic voltammograms (CV) were recorded using a 1 mL electrochemical cell by potential sweeping from an initial potential of −0.1 V to an end-point potential of −0.7 V at different scan rates in a range of 0.01–0.1 V s^−1^.

The preliminary incubation step of diclofenac with SPE/DDAB/CYP2C9 + FAD during 30 min for the saturation of enzyme with substrate was used [30]. For quantification of 4′-hydroxydiclofenac by the square-wave voltammetry (SWV) method, the working electrodes were modified by drop casting 1 μL of the SWCNT TUBALL™ BATT H_2_O dispersion prediluted 10 times with distilled water. The electrodes were kept at room temperature until completely dry. 4-hydroxydiclofenac was registered at the potential of E = 0.12 V. Horizontal mode was used for the analysis of metabolite [32].

Square-wave voltammograms in the 0–1.2 V potential range were recorded at 25 Hz frequency, 40 mV amplitude, and 5 mV potential step. The obtained voltammograms were smoothed and baseline-corrected by the potentiostat software. Naproxen O-demethylation was registered based on formaldehyde formation as described [44].

All experiments were performed in triplicate. The data are presented as average values ± standard deviations (± SD).

## 4. Conclusions

Earlier, the influence of various medications on the activity of cytochrome P450, such as antioxidants (mexidol, taurine) [8,20] and the water-soluble pharmaceutical form of phospholipid nanoparticles [42], was studied. Food-drug and drug–drug interactions remain an important issue in personalized medicine, especially in the context of geriatrics and aging. Cytochrome P450-based electrochemical systems have demonstrated their effectiveness as tools for assessing the activity of this class of enzymes. We have shown that the electrocatalytic activity of CYP2C9 can be significantly enhanced by modeling the electron transfer relay on the electrode by including FAD or FMN as mediators of electrons and low molecular models of reductase and antioxidant molecules. Differences in the electrochemical behavior of diclofenac as a substrate and 4′-hydroxydiclofenac as a metabolite were used for the quantitative electrochemical determination of the catalytic efficiency of electroenzymatic reactions. The data obtained indicate that the addition of coenzyme forms of vitamin B2, namely the flavin cofactor FAD, to the CYP2C9 system activates the catalytic functions of the enzyme and promotes substrate conversion of diclofenac by 148 ± 10% and of naproxen by 120 ± 14%. Based on our previous results [31] and earlier published data [25], we can conclude that flavins, as vitamin B2 predecessors, play a positive role in the metabolic transformation of organic compounds and drugs, catalyzed by the CYP enzyme family. These results showed the relevance of flavins to health and their potential to modulate drug metabolism pathways catalyzed with cytochrome P450s enzymes. The intake of vitamin B2 improves drug metabolism, catalyzed by CYPs. To summarize, for elderly individuals who use non-steroidal anti-inflammatory medications, it may be advantageous to consider prescribing a dietary supplement that contains vitamin B2.

## Data Availability

The data are available on reasonable request from the corresponding author.

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
