# Peer review of "The Activation of Cytochrome P450 2C9 Is Facilitated by the Coenzyme Forms of Vitamin B2"

_molecules, 2025, doi:10.3390/molecules30183673_

Round 1
Reviewer 1 Report
Comments and Suggestions for Authors
In the manuscript, the authors found that the possible of Vitamin B2 in the activation of CYP2C9, and the authors successfully found the improvement of Vitamin B2 in the activation of CYP2C9 based on the electron transfer theory using electrochemical analysis. The manuscript may help understand the function of vitamins in drug medication, especially understand the influence of drug-drug interactions and food-drug interactions in cytochrome P450 enzyme activation. The workload and the creativity of the manuscript are both enough. Therefore, the manuscript is recommended to be accepted by the journal Molecules. Here are some concerns for the authors to improve the quality of the manuscript:
1) When writing the symbol “±” two blank spaces should be added before and after it. The authors should carefully check such errors in the present manuscript including the Abstract, Table, Conclusion sections.
2) In the Keywords section, the first word should be revised as CYP2C9 directedly. And the word ageing should be deleted.
3) The 2nd paragraph of the Introduction section is just one sentence; it reads present no relationship with the 1st and the 3rd Therefore, it is recommended to be deleted.
4) The 1st and the 3rd sentences in the present 4th paragraph of the Introduction section should be supported by references.
5) In the 5th paragraph of the Introduction section, the last sentence read replicated with the 1st sentence of the 6th Therefore, the authors should revise such writing.
6) In Scheme 1, the authors may need to label the chemicals’ name under the chemical structure.
7) In line 96, the authors should clarify the detail author (“XX et al.”).
8) In section 2.1, the font of the 2nd paragraph should adjust.
9) In the 1st paragraph of section 3, many purities of purchased chemicals should be provided.
10) In the last sentences of the 1st and 2nd paragraph of section 3, the website should be deleted. And the authors should provide the city and country information.
11) The author drew 3 schemes to describe the chemicals’ transformation. However, their formats should be revised consistently.
12) As the authors have found such mechanism, the authors should provide a detailed graphic abstract to summary the main discovery using reaction route.
Author Response
Comments 1: When writing the symbol “±” two blank spaces should be added before and after it. The authors should carefully check such errors in the present manuscript including the Abstract, Table, Conclusion sections.
Response 1: We thank the Reviewer 1 for this comment. We added the symbol “±” with two blank spaces throughout the text including the Abstract, Table, Conclusion sections.
Comments 2: In the Keywords section, the first word should be revised as CYP2C9 directedly. And the word ageing should be deleted.
Response 2: We corrected the Keywords section in accordance with Reviewer 1 recommendation.
Comments 3: The 2ndparagraph of the Introduction section is just one sentence; it reads present no relationship with the 1st and the 3rd Therefore, it is recommended to be deleted.
Response 3: The 2ndparagraph of the Introduction section is just one sentence; it reads present no relationship with the 1st and the 3rd Therefore, it is recommended to be deleted.
Comments 4: The 1st and the 3rd sentences in the present 4th paragraph of the Introduction section should be supported by references.
Response 4: We added references for the 1st and the 3rd sentences in the present 4th paragraph (p. 2 of manuscript)
[6] Wauthier, V., Verbeeck, R. K., Calderon, P. B. The effect of ageing on cytochrome P450 enzymes: consequences for drug biotransformation in the elderly. Current medicinal chemistry, 2007, 14(7), 745–757. DOI:10.2174/092986707780090981
Comments 5: In the 5th paragraph of the Introduction section, the last sentence read replicated with the 1st sentence of the 6th Therefore, the authors should revise such writing.
Response 5: We thank Reviewer 1 for this suggestion. We revised these parts of the Introduction section in accordance with Reviewer 1 recommendation.
Comments 6: In Scheme 1, the authors may need to label the chemicals’ name under the chemical structure.
Response 6: We thank Reviewer 1 for this suggestion. In Scheme 1, we labeled the chemicals’ name under the chemical structure (p. 2 of manuscript).
Comments 7: In line 96, the authors should clarify the detail author (“XX et al.”).
Response 7: We added the author details as Wang et al. (p. 3).
Comments 8: In section 2.1, the font of the 2ndparagraph should adjust.
Response 8: We corrected this mistake and changed the font from 9 to 10.
Comments 9: In the 1stparagraph of section 3, many purities of purchased chemicals should be provided.
Response 9: We thank Reviewer 1 for this suggestion. We added the purities of purchased chemicals in Experimental Section (p. 13 of manuscript).
Comments 10: In the last sentences of the 1stand 2nd paragraph of section 3, the website should be deleted. And the authors should provide the city and country information.
Response 10: We added the website, the city and country information in Section 3 in accordance with Reviewer 1 recommendation (p. 14 of manuscript).
Comments 11: The author drew 3 schemes to describe the chemicals’ transformation. However, their formats should be revised consistently.
Response 11:
In Scheme 1, we labeled the chemicals’ name under the chemical structure (p. 2 of manuscript).
In Scheme 2 the CYP2C9-dependent diclofenac metabolism to 4′-hydroxydiclofenac and in Scheme 3 metabolic transformation of naproxen catalyzed by CYP2C9. We described the enzymatic transformation of these drugs with CYP2C9 in accordance with known mechanism of these chemical reactions.
Comments 12: As the authors have found such mechanism, the authors should provide a detailed graphic abstract to summary the main discovery using reaction route.
Response 12: We prepared a detailed graphic abstract to summary the main discovery of our paper.
Reviewer 2 Report
Comments and Suggestions for Authors
In this manuscript Koroleva and collaborators investigated the stimulation of the cytochrome P450 2C9 catalytic activity by riboflavin, FAD and FMN. This is an interesting manuscript but several potential pitfalls due to the lack of proper controls and blanks in almost all experiments. Reduction/oxidation of vitamin B2 and its derivates could lead to the generation of redox species leading to oxidative catalysis of the substrate that is not associated with cytochrome P450 2C9 function. Under these conditions, the manuscript cannot be accepted for publication but authors are encouraged to resubmit once all questions have been addressed and the manuscript has been completed with the appropriate controls and blanks.
Major corrections
1) The authors indicated in lines 153-159 “A typical shoulder at 450-475 nm, indicating the presence of flavins, as depicts Fig. 1A, with a concentration dependence on the flavin content (Fig.1B). The complex of CYP2C9+FAD demonstrates a hyperchromic effect with a bathochromic shift of 10 nm, confirming the interaction of these components. These results are comparable to those obtained in [27].”
In Fig. 1A, authors should plot the FAD spectrum alone using the same concentration as in the CYP2C9+FAD experiments. The authors should compare the spectra and the absorbance titration with free FAD to the same experiments performed in the absence of CYP2C9. Once performed, the authors should describe the changes in free FAD spectra versus those of the FAD:CYP2C9 complex. Also, they should indicate the wavelengths at which they found a bathochromic shift of 10 nm.
Also regarding Fig. 1B, the authors should plot the absorbance dependence on FAD concentration in experiments performed with FAD alone (in the absence of CYP2C9) and add this data to Fig. 1B. This is a control experiment that will let the authors know if complexation accounts, not overestimating the results. This correction might also lead to changes in the calculated Kd value that might need to be corrected. It was also noted that the authors did not indicate the absorbance wavelength used in the figure 1B and its caption.
The authors commented on complex formation between CYP2C9 and FAD but what is the measured stoichiometry for the complex formation? Authors showed the FAD:CYP2C9 UV-vis spectra at 1:10 ratio but what is the rationale for choosing this ratio? and why the authors did not complete the titration to saturation? The authors should show a saturation dependence on absorbance, suggesting a saturation point in the formation of a potential complex, if it accounts.
2) Page 185-196. Authors should indicate in the main text the reference electrode used in the calculation of the redox potential. They indicated in the Methods that they used an Ag/AgCl. This will help convert the values to NHE since most of the studies with biological systems use NHE for comparison purposes with other molecules.
In voltammetry experiments performed with the FAD:CYP2C9 complex, the authors should compare these voltammograms to those of free FAD and buffer alone, since FAD is redox active. It may occur that CYP2C9 shifts its redox potential after binding to FAD, but to detect these change authors need to plot the voltammogram versus buffer in the absence of FAD. The authors should describe possible changes in the formal potential regarding free CYP2C9 and FAD but also after complex formation. Might it happen that reported ΔEp 0.168 ± 0.006 V is associated with observed peaks registered with free FAD? verify.
3) Voltammograms showed in Fig. 3 are not indicative of diclofenac oxidation. They were obtained in the presence of oxygen and they correspond to the signal of oxygen, with only minor changes under the reported conditions. These voltammograms can be compared to those obtained on a silver working electrode with an Ag/AgCl reference, the ORR peak usually appears near -0.4 V vs Ag/AgCl in oxygen-saturated solutions. The use of this signal peak may have been intended to monitor the coupling efficiency in the catalysis of cytochrome P450s in other studies, but it is not very useful for monitoring the catalytic activity of CYP2C9. Indeed, an increase in the oxygen signal intensity might be generated by oxidation of the electrodes or by changes in each condition, not necessarily due to generation of 4′-hydroxydiclofenac or just by degradation of diclofenac in different conditions not associated with the presence of CYP2C9.
Some research groups have studied the oxidation of diclofenac and reported oxidation peaks at 0.87 V and 1.27 V (vs Ag/AgCl) which should be used to properly describe the catalytic activity for the samples and conditions tested.
4) Control experiments with free FAD and FMN are missing in Figure 4 and 5. Free FAD and FMN can generate hydrogen peroxide through oxidation/reduction cycles and non-enzymatic generation of 4’-hydrooxydiclofenac by diclofenac oxidation has been described in the presence of hydrogen peroxide. Please perform experiments using FAD and FMN in the presence of 4’-hydrooxydiclofenac or diclofenac in the absence of CYP2C9.
5) A better description of the oxidation processes accounting in the electrodes in the different tested conditions shown in Fig. 6 is required. Also, the peaks might not be specifically associated with Naproxen. This oxidation reaction in each condition (using appropiate controls such as free FAD or FMN in non-reconstructed complexes with CYP2C9 ) should be better characterized before publication of this manuscript.
Minor corrections
- Please indicate the material and metal used in the construction of the screen-printed electrode in the main text and results section.
- Lines 317-321: Units for Vmax are missing.
- The title for Table 4 is missing.
- Please improve the quality of figures and increase the size of the font size for x and y -axes.
Author Response
Major corrections
Comments 1: The authors indicated in lines 153-159 “A typical shoulder at 450-475 nm, indicating the presence of flavins, as depicts Fig. 1A, with a concentration dependence on the flavin content (Fig.1B). The complex of CYP2C9+FAD demonstrates a hyperchromic effect with a bathochromic shift of 10 nm, confirming the interaction of these components. These results are comparable to those obtained in [27].”
In Fig. 1A, authors should plot the FAD spectrum alone using the same concentration as in the CYP2C9+FAD experiments. The authors should compare the spectra and the absorbance titration with free FAD to the same experiments performed in the absence of CYP2C9. Once performed, the authors should describe the changes in free FAD spectra versus those of the FAD:CYP2C9 complex. Also, they should indicate the wavelengths at which they found a bathochromic shift of 10 nm.
Also regarding Fig. 1B, the authors should plot the absorbance dependence on FAD concentration in experiments performed with FAD alone (in the absence of CYP2C9) and add this data to Fig. 1B. This is a control experiment that will let the authors know if complexation accounts, not overestimating the results. This correction might also lead to changes in the calculated Kd value that might need to be corrected. It was also noted that the authors did not indicate the absorbance wavelength used in the figure 1B and its caption.
The authors commented on complex formation between CYP2C9 and FAD but what is the measured stoichiometry for the complex formation? Authors showed the FAD:CYP2C9 UV-vis spectra at 1:10 ratio but what is the rationale for choosing this ratio? and why the authors did not complete the titration to saturation? The authors should show a saturation dependence on absorbance, suggesting a saturation point in the formation of a potential complex, if it accounts.
Response 1: We thank Reviewer 2 for these comments and advises. In our study, we investigated the binding of FAD with CYP2C9. We assumed that the shoulder at 450 nm reflects the binding events, as was shown earlier (Bioelectrochemistry, 149 (2023) 108277). Absorption spectra of CYP2C9 have the Soret peak characteristic of heme proteins (400-412 nm), and complex of CYP2C9+FAD additionally demonstrate a wide shoulder at 450-475 nm region indicating the presence of flavins. Docking of riboflavin, FMN and FAD to the surface of cytochrome P450 3A4 showed that binding site is located in the cavity on the proximal side of hemeprotein and not overlapped with substrate binding site. This region is known as binding site of redox partners of cytochromes P450, for example, cytochrome b5 that participates in the regulation of cytochrome P450 activity and electron transport. Flavin fragment of riboflavin, FMN and FAD located in similar region with good overlap. Based on the similarity of binding sites for flavin fragment we can assume the same tendencies for CYP2C9.
- Biol. Chem. 288 (30) (2013) 22080–22095, https://doi.org/10.1074/jbc.M112.448225; J. Biol. Chem. 273 (27) (1998) 17036–17049; Biochemistry (Mosc.) 66 (2001) 667–681, https://doi.org/10.1023/A:1010215516226; Arch. Biochem. Biophys, 354 (1998) 133-138. https://doi.org/10.1006/abbi.1998.0628, ChemBioChem. 21 (2020) 2297-2305. https://doi.org/10.1002/cbic.202000071; Biochem. Biophys. Res. Commun. 1994, 203, 1745–1749. DOI:10.1006/bbrc.1994.2388.
We added the additional description of Fig. 1A (p. 6 of manuscript).
Comments 2: Page 185-196. Authors should indicate in the main text the reference electrode used in the calculation of the redox potential. They indicated in the Methods that they used an Ag/AgCl. This will help convert the values to NHE since most of the studies with biological systems use NHE for comparison purposes with other molecules.
In voltammetry experiments performed with the FAD:CYP2C9 complex, the authors should compare these voltammograms to those of free FAD and buffer alone, since FAD is redox active. It may occur that CYP2C9 shifts its redox potential after binding to FAD, but to detect these change authors need to plot the voltammogram versus buffer in the absence of FAD. The authors should describe possible changes in the formal potential regarding free CYP2C9 and FAD but also after complex formation. Might it happen that reported ΔEp 0.168 ± 0.006 V is associated with observed peaks registered with free FAD? verify.
Response 2: We thank Reviewer 2 for this comment. On p. 15 of experimental section we mentioned that screen-printed electrodes (SPE) consists of the graphite working electrode (surface area 0.0314 cm2), an auxiliary electrode and a silver/silver chloride pseudo reference electrode. All potentials were given vs Ag/AgCl electrode (p.7 and 17 of manuscript). We added this experimental detail in the Results and Discussion part (p. 7 of manuscript).
We also added Fig. 3B depicted cyclic voltammograms of SPE/DDAB and cyclic voltammograms of SPE/DDAB/FAD. Reported shift of potential for ΔEp 0.168 ± 0.006 V is associated with complex formation of CYP2C9 and FAD molecule and the influence of Flavin on the reduction potential of heme.
Comments 3: Voltammograms showed in Fig. 3 are not indicative of diclofenac oxidation. They were obtained in the presence of oxygen and they correspond to the signal of oxygen, with only minor changes under the reported conditions. These voltammograms can be compared to those obtained on a silver working electrode with an Ag/AgCl reference, the ORR peak usually appears near -0.4 V vs Ag/AgCl in oxygen-saturated solutions. The use of this signal peak may have been intended to monitor the coupling efficiency in the catalysis of cytochrome P450s in other studies, but it is not very useful for monitoring the catalytic activity of CYP2C9. Indeed, an increase in the oxygen signal intensity might be generated by oxidation of the electrodes or by changes in each condition, not necessarily due to generation of 4′-hydroxydiclofenac or just by degradation of diclofenac in different conditions not associated with the presence of CYP2C9.
Some research groups have studied the oxidation of diclofenac and reported oxidation peaks at 0.87 V and 1.27 V (vs Ag/AgCl) which should be used to properly describe the catalytic activity for the samples and conditions tested.
Response 3: Figure 3A demonstrated non-catalytic reduction current in aerobic conditions for CYP2C9 and catalytic current in the presence of diclofenac (red lines) and non-catalytic reduction current for CYP2C9+FAD and catalytic current in the presence of diclofenac (blue lines). The ratio Icat/Ired reflects the repartition of the electron flow between substrate and oxygen and can serve an indicator of coupling efficiency in the catalysis of cytochrome P450s. For the estimation of product formed during CYP2C9-dependent catalysis, we used original approach elaborated in our laboratory earlier. Our all-electrochemical approach is based on the estimation of the electrochemical oxidation of a metabolic product using a SPE, modified with single-walled carbon nanotubes (SPE/SWCNT) (Electrocatalysis. 2022, 13, 630-640. DOI:10.1007/s12678-022-00753-3.)
Comments 4: Control experiments with free FAD and FMN are missing in Figure 4 and 5. Free FAD and FMN can generate hydrogen peroxide through oxidation/reduction cycles and non-enzymatic generation of 4’-hydrooxydiclofenac by diclofenac oxidation has been described in the presence of hydrogen peroxide. Please perform experiments using FAD and FMN in the presence of 4’-hydrooxydiclofenac or diclofenac in the absence of CYP2C9.
Response 4: Earlier, we have shown that in our condition there is not observed reduction peak of FAD and FMN at used concentration (210 μM) at investigated potentials from -0.1 to -0.7 V at SPE. In addition, in our publication (Bioelectrochemistry, 149 (2023) 108277) we have showed that in the presence of riboflavin hydrogen peroxide formation is not increased during electrolysis comparable to system without riboflavin. Our experimental data are in accordance with previous publications concerning the influence of riboflavin on the catalytic efficiency of CYP enzymes.
Comments 5: A better description of the oxidation processes accounting in the electrodes in the different tested conditions shown in Fig. 6 is required. Also, the peaks might not be specifically associated with Naproxen. This oxidation reaction in each condition (using appropiate controls such as free FAD or FMN in non-reconstructed complexes with CYP2C9) should be better characterized before publication of this manuscript.
Response 5: Figure 6 is demonstrated the difference between reduction currents of CYP2C9 and CYP2C9+FAD in the presence of naproxen in aerobic conditions, and CYP2C9 and CYP2C9+FAD in aerobic conditions, respectively.
CYP2C9-dependent and CYP2C9+FAD catalytic processes of naproxen N-demethylation are presented in Table 4. The catalytic current of naproxen (Icat max) for CYP2C9+FAD complex increased from -0.34±0.07 µA to -4.22±0.08 µA in comparison with CYP2C9. For the estimation of effectivity of catalysis we used the measuring procedure for formaldehyde determination in accordance with well-known approach (Biochem. J. 1953, 55, 416–421, Bioelectrochemistry. 2021, 138, 107729). Fig. 6 depicted the difference between the non-catalytic current for CYP2C9 or CYP2C9+FAD and catalytic current in the presence of substrate naproxen. As can be seen from Fig. 6, complex CYP2C9 +FAD demonstrates effective metabolic transformation of drug with more pronounced catalytic current.
Minor corrections
Comments 1: Please indicate the material and metal used in the construction of the screen-printed electrode in the main text and results section.
Response 1: We added the detailed description of materials used for the electrodes preparations (p. 13 of manuscript)
Comments 2: Lines 317-321: Units for Vmax are missing.
Response 2: We added M/min as units for Vmax
Comments 3: The title for Table 4 is missing.
Response 3: We added the title for Table 4 as Table 4. Electrochemical and electrocatalytical parameters of SPE/DDAB/CYP2C9 and SPE/DDAB/CYP2C9+FAD under aerobic conditions and in the presence of 100 µM naproxen.
Comments 4: Please improve the quality of figures and increase the size of the font size for x and y -axes.
Response 4: We significantly improved the quality of figures and increase the size of the font size for x and y –axes (Figs. 1- 6).
We thank the reviewer for a thorough analysis of our manuscript and have made corrections in accordance with reviewer comments.
Round 2
Reviewer 1 Report
Comments and Suggestions for Authors
The 2nd edition of the manuscript has been improved. And the present version of the manuscript is recommended to be accepted by the journal Molecules. Congratulations to the authors.
Reviewer 2 Report
Comments and Suggestions for Authors
Thank you for answering my questions and for experimentally proving the points that were raised during the review process. The manuscript has been significally improved and is now ready for publication.